# Barely Random Algorithms
# and Collective Metrical Task Systems

**Romain Cosson**
Inria, Paris
romain.cosson@inria.fr

**Laurent Massoulié**
Inria, Paris
laurent.massoulie@inria.fr

## Abstract

We consider metrical task systems on general metric spaces with $n$ points, and show that any fully randomized algorithm can be turned into a randomized algorithm that uses only $2\log n$ random bits, and achieves the same competitive ratio up to a factor 2. This provides the first order-optimal *barely random* algorithms for metrical task systems, i.e. which use a number of random bits that does not depend on the number of requests addressed to the system. We discuss implications on various aspects of online decision making such as: distributed systems, advice complexity and transaction costs, suggesting broad applicability. We put forward an equivalent view that we call *collective metrical task systems* where $k$ agents in a metrical task system team up, and suffer the average cost paid by each agent. Our results imply that such team can be $O(\log^2 n)$-competitive as soon as $k \geq n^2$. In comparison, a single agent is always $\Omega(n)$-competitive.

## 1 Introduction

Recent progress on the competitive analysis of important online problems, such as the $k$-server problem, metrical service systems and metrical task systems have relied on the introduction of continuous optimization methods, such as the online primal-dual framework (see e.g. the book of [17]), and more recently, the online mirror descent framework [13–15, 5]. By design, these methods assume that the online algorithm is provided with an infinite number of random bits. In this paper, we address the question of whether this requirement is inherent to the online problem, or specific to the methods at hand, by focusing on the influential example of metrical task systems [46, 10, 33, 14, 22].

To answer this question, we study the notion of barely random algorithm, introduced by [44] in the early days of competitive analysis of online algorithms: "We call an algorithm that uses a bounded number of random bits regardless of the number of requests *barely random*". In the case of metrical task systems, we will observe that this notion is particularly fruitful, as it connects various aspects of online decision making such as collective systems, switching costs, and advice complexity.

Metrical Task Systems (MTS) is a central problem in online algorithms and online learning [43, 19, 28, 35, 18, 26] which recently attracted the interest of the community working on learning-augmented algorithms [1, 20, 3, 21, 4]. In essence, the problem is similar to the classical learning setting of 'prediction with expert advice' [41], with the key difference that there is a cost associated to switching between experts. We now describe the setting of metrical task systems [12] in a way that highlights the role played by the source of randomness.

*Metrical Task Systems.* The problem is defined on a metric space $\mathcal{X} = (X, d)$ where $X$ is a finite set of cardinality $|X| = n$. The input is a sequence of cost vectors $\boldsymbol{c}(\cdot) = (\boldsymbol{c}(t))_{t\in\mathbb{N}} \in (\mathbb{R}_+^X)^\mathbb{N}$ and the output is a sequence of states $\rho(\cdot) = (\rho(t))_{t\in\mathbb{N}} \in X^\mathbb{N}$. The cost associated to each time $t \in \mathbb{N}$ is the sum of the movement cost $d(\rho(t-1), \rho(t))$ and the service cost $c_{\rho(t)}(t)$. The cost over all time steps

38th Conference on Neural Information Processing Systems (NeurIPS 2024).

therefore writes as follows:

$$\text{Cost}(\rho(\cdot), \boldsymbol{c}(\cdot)) = \sum_{t \geq 1} d(\rho(t-1), \rho(t)) + c_{\rho(t)}(t).$$

The offline benchmark, denoted by $\text{OPT}(\boldsymbol{c}(\cdot))$, is defined as the smallest achievable cost over all possible sequences of states, i.e. $\text{OPT}(\boldsymbol{c}(\cdot)) = \inf_{\rho(\cdot) \in \mathcal{X}^{\mathbb{N}}} \text{Cost}(\rho(\cdot), \boldsymbol{c}(\cdot))$. An online algorithm $\mathcal{A}$ is a method to define a time-consistent trajectory, i.e. such that the state at some time depends only on current and past information. It may use some source of randomness $\boldsymbol{s}$ (seed) which is sampled from a known probability distribution $\boldsymbol{s} \sim \mathcal{D}$. Formally, the sequence of states $\rho_{\mathcal{A}}(\cdot)$ defined by algorithm $\mathcal{A}$ can be computed by $\forall t : \rho_{\mathcal{A}}(t) = \mathcal{A}(\boldsymbol{c}(1), \ldots, \boldsymbol{c}(t), \boldsymbol{s})$. The algorithm $\mathcal{A}$ is said to be $\alpha$-competitive, for some $\alpha \in \mathbb{R}$, if it satisfies $\mathbb{E}_{\boldsymbol{s} \sim \mathcal{D}}(\text{Cost}(\rho_{\mathcal{A}}(\cdot), \boldsymbol{c}(\cdot))) \leq \alpha \times \text{OPT}(\boldsymbol{c}(\cdot)) + \beta$ for some fixed $\beta \in \mathbb{R}$ and for any $\boldsymbol{c}(\cdot) \in (\mathbb{R}_+^X)^{\mathbb{N}}$. In this paper, we are interested in the way the competitive ratio scales with the source of randomness $\mathcal{D}$. When $\mathcal{D} = \mathcal{U}(\{0,1\}^b)$, or equivalently when $\mathcal{D} = \mathcal{U}(\{1, \ldots, 2^b\})$, we say that the algorithm is provided with $b$ random bits. More generally, we consider the case where $\mathcal{D} = \mathcal{U}(\{1, \ldots, k\})$ for arbitrary $k \in \mathbb{N}$. In one extreme, the case of $k = 1$ is known as the *deterministic* variant of the problem and is well understood since the inception of metrical task systems. In the other extreme, the case of $k = +\infty$ (more formally, $\mathcal{D} = \mathcal{U}(\{0,1\}^{\mathbb{N}})$) is known as the *randomized* variant of the problem and was solved more recently. Any other value of $k \in \mathbb{N}$ defines the *k-barely random* variant, which is the focus of this paper. For reasons that will now become apparent, we will also refer to this setting as the *k-collective* variant.

*Collective algorithm for metrical task systems.* Consider a team of $k \in \mathbb{N}$ agents that are collectively confronted to a metrical task system. We can denote the positions of the agents by $\rho_1(t), \ldots, \rho_k(t) \in X^k$ and their trajectory is given by $k$ deterministic online algorithms $\mathcal{A}_1, \ldots, \mathcal{A}_k$. The participants are collaborating, in the sense that the cost paid by team is the average cost paid by its members. It is immediate to observe that *k-collective* setting of metrical task systems is exactly the same problem as the *k-barely random* setting defined above. This is because, given a $k$-collective strategy, one can define a $k$-barely random algorithm by using the seed $\boldsymbol{s} \sim \mathcal{U}(\{1, \ldots, k\})$ to sample uniformly at random one strategy to imitate. In the rest of the paper, we will thus use the terms *collective* and *barely random* indistinctively. The collective presentation reflects a variety of real-world situations, where a finite team of agent is confronted to an adversarial environment. A simple biological illustration is in foraging, e.g. when a colony of $k$ ants is tasked to collectively gather a large amount of food (modeled by the service costs) from $n$ locations, while limiting the collective energy spent (modeled by the movement costs) [32, 36].

## 1.1 Main contributions

Our main technical result is the following,

**Theorem 1.1** (Section 3). *For any metric space $\mathcal{X}$ with $n$ points, for any $\epsilon > 0$, for any integer $k \geq n^2/\epsilon$, if there exists a (fully) randomized MTS algorithm that is $\alpha$-competitive, then there exists a $k$-barely random MTS algorithm that is $(1 + \epsilon)\alpha$-competitive.*

The factor 2 announced in the abstract can be readily recovered from this theorem by letting $\epsilon = 1$, which we can assume in the rest of this section for ease of presentation. Theorem 1.1 relies on two techniques: a new equivalence between *barely random* algorithms and *barely fractional* strategies for metrical task systems that is presented in Section 2 ; and a new discrete first-order method that is developed in Section 3.

In light of the $\mathcal{O}(\log^2 n)$-competitive randomized algorithm of Bubeck et al. [14] for metrical task systems, Theorem 1.1 has the following consequence:

**Corollary 1.2** (of Theorem 1.1). *There is a $\mathcal{O}(\log^2 n)$-competitive algorithm for MTS that requires on $\lceil 2\log n \rceil$ random bits on any $n$ point metric.*

We provide in Section 4 a simple lower-bound that almost matches this result. We also provide a tight guarantee in the case of the uniform metric space.

**Proposition 1.3** (Section 4). *Any $\mathcal{O}(\log^2 n)$-competitive algorithm for MTS on an $n$ point metric requires at least $\log n - \mathcal{O}(\log \log n)$ random bits. In the uniform metric space, there is a $\mathcal{O}(\log n)$-competitive algorithm that only requires $\lceil \log n \rceil$ random bits.*

Finally, we observe that the notion of barely random algorithm is closely connected to the notion of advice complexity. Advice complexity was introduced by [29] to measure the information content of online problems. In our context, one could ask the following question: What amount of information about the cost sequence suffices to improve over the $2n - 1$ lower-bound on the deterministic competitive ratio of metrical task systems? Since bits of advice are at least as useful as random bits (see [37] for more details), we have:

**Corollary 1.4** (of Theorem 1.1). *For any $n$ point metric, there exists a deterministic MTS algorithm that is $\mathcal{O}(\log^2 n)$-competitive, using only $\lceil 2 \log n \rceil$ bits of advice.*

This is a clear improvement over the result of [34] which requires an advice of size that scales with the length of the cost sequence.

## 1.2   Other applications

Our method is helpful in any setting where acquiring random bits is costly. In concrete implementations and real-world systems, we also expect that barely fractional configurations (which are the discrete objects manipulated in this paper) are generally much more practical to handle than fully fractional distributions (which are arbitrary real numbers).

Our results also imply that a finite team of deterministic agents can be competitive in a way no single deterministic agent could be. This echoes of the study of intelligence in multi-agent system, with concrete applications, such as dynamic power management [3]. Consider for example a computer with three power modes (*on*, *sleep*, *off*) with a switching cost of 1 between *on* and *sleep* and another switching cost of, say, 5 between *sleep* and *off*. At each time $t$ the request is either that the computer is used and $c(t) = (1, +\infty, +\infty)$ i.e. the computer must be in state *on* ; or the computer is not used and $c(t) = (1, 0.5, 0)$ i.e. the computer would rather be *off* than *sleep*, and *sleep* than *on*. Consider the variant of the problem, where the computer is in fact composed of $k = 9$ components (e.g. screen, CPU cores, etc.) that can each individually switch between one of the three modes (*on*, *sleep*, *off*). The energy spent by the computer is the sum of energy spent by its constituents. Since $n = 3$ and $k > n^2$, our results imply that such system can enjoy the *randomized* competitive ratio, even if the cost sequence $c(\cdot)$ depends on the state of the system (*deterministic* adversary).

We also note that our work has a flavor that is similar to learning-augmented algorithms. Learning-augmented algorithms use predictions coming from a black-box algorithm in the hope to improve over the performance of competitive algorithms when predictions are accurate. In the same spirit, our method in Section 3 uses a fully fractional distribution coming from a black-box algorithm, and tracks it via a proximal method.

## 1.3   Related works

Metrical task systems is a problem introduced by Borodin et al. (1992) [12]. This initial work resolved the deterministic competitive ratio, which is of $2n - 1$ for any metric space with $n$ points; as well as the randomized competitive ratio for the uniform metric space, which is in $\Theta(\log n)$. The randomized competitive ratio for general $n$ point metrics remained open until Bubeck et al. (2021) proposed a $\mathcal{O}(\log^2 n)$-competitive algorithm and [16] obtained a matching lower-bound. These breakthroughs followed a long line of research (see e.g. [46, 10, 33, 22] and references therein).

Metrical task systems is deeply connected (both, in results and methods) to the classical online setting of "learning with expert advice", as observed by [9]. The differences are (1) there is a (movement) cost associated to switching from one expert to another (2) the offline benchmark is the best moving strategy as opposed to the best fixed strategy (3) the goal is to obtain a multiplicative guarantee (competitive ratio) rather than an additive guarantee (regret).

The notion of barely random algorithms was introduced by [44] for the list update problem. In this paper, they present an algorithm that is 1.75-competitive for the list update problem, using exactly $n$ random bits, where $n$ is the length of the list. Such algorithms are further studied in the celebrated book of [11]. Barely random algorithms were later investigated for paging [37]. The notion also had some echo in the literature on scheduling (see [31] and references therein) but it was not applied before to metrical task systems or related problems.

Advice complexity was first studied in the context of metrical task systems in [30]. In this paper, the authors study the setting where $b$ bits of advice are provided at each time-step to a deterministic metrical task systems algorithm, for $b$ of order $\log n$. The total amount of advice they require is therefore of order $B = T \log n$, where $T$ is the number of time-steps. One surprising consequence of our results (Corollary 1.4) is that we present a deterministic algorithm that has a competitive ratio of $\mathcal{O}(\log^2 n)$ using a single advice of total size $B = 2 \log n$, i.e. which is independent of the number of time-steps $T$. We note that advice complexity is connected to learning-augmented algorithms, when the advice is untrusted [2].

Collective algorithms originate from the literature on distributed algorithms. For example, in the problem of collective tree exploration, a team of $k$ agents is tasked to go through all edges of some unknown tree as fast as possible [34]. The connections observed in [24] between this problem and randomized algorithms for metrical service systems provided some inspiration for the present paper. We note that the interpretation of a probability distribution as a continuous swarm of infinitesimally small agents is not novel, see e.g. [6], and is directly connected to the so-called fractional formulation of metrical task systems. The novelty of our methods lies in the rigorous discretization of this fractional view, and leads to applications in collective and barely random algorithms. A surprising aspect of our method is that it does not rely on a tree-embedding of the metric space, but seems naturally adapted to the general setting, thanks to the very general properties coming from optimal transport, such as the Birkhoff-von Neuman theorem. The discretization also relies on a non-trivial (though concise) first-order method, and which could be of interest for future works.

Finally, Section 3 of this paper is reminiscent of first-order optimization methods. Gradient descent of a function $f(\cdot)$ is also known as the explicit Euler method, where one can write its $t$-th iterate as the minimizer of the sum of a first order approximation of $f(\cdot)$ at the $t-1$-th iterate and a quadratic cost. A closely related first-order method is the implicit Euler method where the $t$-th iterate is defined by,

$$\boldsymbol{x}(t) = \arg\min_{\boldsymbol{x}} f(\boldsymbol{x}) + ||\boldsymbol{x} - \boldsymbol{x}(t-1)||^2,$$

which is also known as the proximal operator [42]. Our Equation (1) is reminiscent of this operator. One key difference is that we replace the squared norm by an optimal transport cost, and that the offline objective function $f(\cdot)$ is replaced by an online potential, like in the work function algorithm [38]. Note that quite recently, the notion of Wasserstein proximal operator has appeared as a central object of study in the context of partial differential equations (see e.g. [45]). A key difference with our Equation (1) is that we consider the Wasserstein-1 metric (aka, the earthmover distance) and not the Wasserstein-2 metric.

### 1.4 Notations, definitions and preliminaries

In the following, $\mathcal{X} = (X, d)$ is a finite metric space, i.e. $X$ is a finite set and the distance $d$ is positive, symmetric and satisfies the triangle inequality. We denote by $n$ the cardinality of $X$.

We denote by $\mathcal{P}(\mathcal{X})$ the set of all distributions on $\mathcal{X}$. For any two such distributions $\boldsymbol{x}, \boldsymbol{x}' \in \mathcal{P}(\mathcal{X})$, we call the optimal transport cost from $\boldsymbol{x}$ to $\boldsymbol{x}'$ and we denote by $\mathrm{OT}(\boldsymbol{x}, \boldsymbol{x}')$ the quantity,

$$\mathrm{OT}(\boldsymbol{x}, \boldsymbol{x}') = \inf_{\pi \in \Gamma(\boldsymbol{x}, \boldsymbol{x}')} \sum_{\rho, \rho'} \pi(\rho, \rho') d(\rho, \rho')$$

where $\Gamma(\boldsymbol{x}, \boldsymbol{x}') = \{\pi \in [0,1]^{\mathcal{X} \times \mathcal{X}} : \forall \rho, \sum_{\rho' \in \mathcal{X}} \pi(\rho, \rho') = x_\rho \text{ and } \forall \rho', \sum_\rho \pi(\rho, \rho') = x'_{\rho'}\}$ is the set of couplings from $\boldsymbol{x}$ to $\boldsymbol{x}'$. Classically, $\Gamma(\boldsymbol{x}, \boldsymbol{x}')$ can be viewed as the probability distributions on $\mathcal{X} \times \mathcal{X}$ with their marginals respectively equal to $\boldsymbol{x}$ and $\boldsymbol{x}'$. The optimal transport cost (also known as the Wasserstein distance) defines a distance between probability distributions on $\mathcal{X}$. In particular, it satisfies positivity, symmetry and the triangle inequality.

For some point $\rho \in \mathcal{X}$, we denote by $\boldsymbol{e}_\rho \in \mathcal{P}(\mathcal{X})$ the probability distribution on $\mathcal{X}$ that has all of its mass on $\rho$. Note that $\boldsymbol{e}_\rho$ can alternatively be seen as a vector of $\mathbb{R}^{\mathcal{X}}$. Denoting by $\otimes$ the Kronecker product (aka the outer product) between vectors of $\mathbb{R}^{\mathcal{X}}$, we have, for any $\rho, \rho' \in \mathcal{X}$ that $\boldsymbol{e}_\rho \otimes \boldsymbol{e}_{\rho'}$ is an optimal coupling from $\boldsymbol{e}_\rho$ to $\boldsymbol{e}_{\rho'}$

For an arbitrary constant $k \in \mathbb{N}$ we denote by $\mathcal{P}_k(\mathcal{X})$ the set of all distributions on $\mathcal{X}$ that only take their values in $\{0, 1/k, \ldots, 1\}$. Such distributions will sometimes be called 'configurations' to distinguish them from their continuous counterparts. For any two configurations $\boldsymbol{x}, \boldsymbol{x}' \in \mathcal{P}_k(\mathcal{X})$,

there exists an associated optimal coupling $\pi$ that only takes its values in $\{0, 1/k, \ldots, 1\}$, i.e. which follows the discrete formulation of optimal transport by Monge. We call such a (discrete) coupling, an optimal transport plan. The existence of this optimal transport plan be seen as a consequence of the Birkhoff-von Neumann theorem which states that the extreme points of the polytope of doubly stochastic matrices are permutation matrices [8]. Indeed a $k \times k$ doubly stochastic matrix naturally induces a coupling between two $k$-barely fractional configurations, with values in $\{0, 1/k, \ldots, 1\}$ if it is a permutation matrix. The result then follows from the linearity of the objective.

For an integer $k \in \mathbb{N}$, we denote by $\mathcal{U}(\{1, \ldots, k\})$ the uniform probability distribution over $\{1, \ldots, k\}$. For an arbitrary distribution $\mathcal{D}$, we denote by $\mathbb{E}_{s \sim \mathcal{D}}(\cdot)$ the expectation when the variable $s$ is sampled following $\mathcal{D}$. All logarithms, denoted by $\log(\cdot)$, are in base 2.

**Paper outline**  Section 2 provides an equivalence between $k$-collective strategies and $k$-barely fractional strategies, which is used in Section 3 to prove Theorem 1.1. Section 4 provides additional discussions, including the lower-bound as well as refined guarantees for the uniform metric space. The conclusion highlights some open directions towards making further connections between online algorithms and distributed/collective systems.

## 2  Barely fractional strategies for metrical task systems

In the introduction, we gave the classical definition of metrical task systems [12], while highlighting the role played by the source of randomness $\mathcal{D}$. In the literature, a variant called the *fractional* formulation is known to be equivalent to the problem when the algorithm is allowed an infinite number of random bits (i.e. when $\mathcal{D} = \mathcal{U}(\{0, 1\}^{\mathbb{N}})$). We start by recalling this fractional formulation and the equivalence in Section 2.1. We then introduce the notion of $k$-*barely fractional* strategies and we show their relevance to the aforementioned $k$-barely random setting in Section 2.2. In contrast with previous results, this equivalence relies on the Birkhoff-von Neumann theorem.

### 2.1  (Fully) fractional strategies for metrical task systems

*Fractional Metrical Task System.* In the fractional variant of metrical task systems, at any instant $t$, a fractional strategy denoted by $\boldsymbol{x}(\cdot)$ maintains a distribution over the states, that only depends on the information available before time $t$, i.e. $\boldsymbol{x}(\boldsymbol{c}(1), \ldots, \boldsymbol{c}(t)) \in \mathcal{P}(\mathcal{X})$, which shall be denoted by $\boldsymbol{x}(t)$ hereafter. The cost paid by the strategy at time $t$ is defined as the sum of a transport cost $\mathrm{OT}(\boldsymbol{x}(t-1), \boldsymbol{x}(t))$ and a service cost $\sum_{\rho \in \mathcal{X}} x_\rho(t) c_\rho(t)$. We denote by $\mathrm{Cost}(\boldsymbol{x}(\cdot), \boldsymbol{c}(\cdot))$ the total cost associated to the strategy, i.e.

$$\mathrm{Cost}(\boldsymbol{x}(\cdot), \boldsymbol{c}(\cdot)) = \sum_{t \geq 1} \mathrm{OT}(\boldsymbol{x}(t-1), \boldsymbol{x}(t)) + \sum_{\rho \in \mathcal{X}} x_\rho(t) c_\rho(t).$$

The interest of this fractional formulation comes from the following reduction, which is classical in the literature (see e.g. [14]).

**Proposition 2.1.** *For any fractional strategy for metrical task systems, there is a (fully) randomized algorithm that achieves the same cost, and reciprocally.*

*Proof.* Consider a randomized algorithm $\mathcal{A}(\cdot, s)$ with $s \sim \mathcal{U}(\{0, 1\}^{\mathbb{N}})$ for metrical task systems and define $\boldsymbol{x}(\cdot)$ at time $t$ by setting $\forall \rho \in X : x_\rho(t) = \mathbb{P}_s(\mathcal{A}(\boldsymbol{c}(1), \ldots, \boldsymbol{c}(t), s) = \rho)$. It is clear from this definition that the service cost of the fractional strategy equals the expected service cost of the randomized algorithm. Also observe that the movement cost of the fractional strategy, which equals $\mathrm{OT}(\boldsymbol{x}(t-1), \boldsymbol{x}(t))$ at time $t$, must be less than the expected movement cost of the randomized algorithm which uses a (possibly sub-optimal) coupling between the two consecutive distributions.

Reciprocally, assume that we are given a fractional strategy $\boldsymbol{x}(\cdot) \in \mathcal{P}(\mathcal{X})^{\mathbb{N}}$. We can design a randomized algorithm $\mathcal{A}(\cdot, s)$ which relies on the random seed $s \sim \mathcal{D}(\{0, 1\}^{\mathbb{N}})$. Observe that we can make the seemingly stronger (but in fact equivalent, because $\{0, 1\}^{\mathbb{N}}$ is uncountable) assumption that $\mathcal{D} = \mathcal{U}([0, 1]^{\mathbb{N}})$. This reformulation allows to have access to one fresh (independent) real sampled uniformly at random from $[0, 1]$ at each time-step $t \in \mathbb{N}$, which we denote by $s(t) \in [0, 1]$. We assume that at time $t - 1$, the algorithm is in some state $\mathcal{A}(\boldsymbol{c}(1), \ldots, \boldsymbol{c}(t-1), s) = \rho(t-1)$. At time $t$, the algorithm considers the optimal transport plan $\pi$ associated to $\mathrm{OT}(\boldsymbol{x}(t-1), \boldsymbol{x}(t))$ and samples

the state $\rho(t)$ following the probability distribution proportional to $(\pi(\rho(t-1), \rho))_{\rho \in \mathcal{X}}$. This can be done by inversion sampling using only the random sample $s(t) \in \mathcal{U}([0,1])$. It is clear by induction that the distribution of $\mathcal{A}(\boldsymbol{c}(1), \ldots, \boldsymbol{c}(t), \boldsymbol{s}) = \rho(t)$ with this randomized algorithm follows exactly $\boldsymbol{x}(t)$. Further, its expected movement cost $\mathbb{E}_{\boldsymbol{s}}(d(\rho(t-1), \rho(t)))$ is exactly equal (by definition) to the transport cost $\mathrm{OT}(\boldsymbol{x}(t-1), \boldsymbol{x}(t))$. $\qquad \square$

## 2.2 Barely fractional strategies for metrical task systems

*Barely Fractional Metrical Task System.* We now introduce a discretization of the fractional formulation of metrical task systems. This will define a $k$-barely fractional strategy $\boldsymbol{x}(\cdot)$. The definitions are the same as in the fully fractional variant above, except that at all times, the distribution $\boldsymbol{x}(t) \in \mathcal{P}(\mathcal{X})$ is constrained to belong to the set $\mathcal{P}_k(\mathcal{X})$ of distributions taking values in $\{0, 1/k, \ldots, 1\}$ (see the notations section). We now show that this formulation enjoys an equivalence with the barely random variant of metrical task systems, when the source of randomness is limited to $\mathcal{D} = \mathcal{U}(\{1, \ldots, k\})$.

**Proposition 2.2.** *For any $k$-barely fractional strategy for metrical task system, there is a $k$-barely random algorithm that acheives the same cost, and reciprocally.*

*Proof.* We assume that we are given a $k$-barely random algorithm $\mathcal{A}$ and define as above the fractional strategy $\boldsymbol{x}(\cdot)$ at time $t$ by $\forall \rho \in X : x_\rho(t) = \mathbb{P}_{\boldsymbol{s}}(\mathcal{A}(\boldsymbol{c}(1), \ldots, \boldsymbol{c}(t), \boldsymbol{s}) = \rho)$. It remains true (see the argument of Section 2.1) that this fractional strategy has a smaller service and movement cost than the randomized algorithm. We also observe that since $\boldsymbol{s} \sim \mathcal{D} = \mathcal{U}(\{0, 1/k, \ldots, 1\})$, we have $\boldsymbol{x}(t) \in \mathcal{P}_k(\mathcal{X})$. Therefore, $\boldsymbol{x}(\cdot)$ is a valid $k$-barely fractional strategy with a smaller cost than $\mathcal{A}$.

The reverse reduction uses an ingredient from the theory of optimal transport: the Birkhoff-von Neumann theorem, which states that the optimal transport between $\boldsymbol{x}, \boldsymbol{x}' \in \mathcal{P}_k(\mathcal{X})$ always admits an optimal coupling $\pi$ taking its values in $\{0, 1/k, \ldots, 1\}$, that we call an optimal transport plan (see notations and preliminaries section). Given a barely fractional strategy $\boldsymbol{x}(\cdot)$, we define a $k$-collective algorithm for which the population distribution will always match $\boldsymbol{x}(t) \in \mathcal{P}_k(\mathcal{X})$. The result then follows from the equivalence between collective and barely random algorithms discussed in the introduction. We initially distribute the $k$ agents following $\boldsymbol{x}(0)$. Now, we assume as our induction hypothesis that the team at time $t-1$ is distributed exactly following $\boldsymbol{x}(t-1)$ and we show that we can deterministically redistribute the members of the team to follow $\boldsymbol{x}(t)$, while paying a moment cost of $\mathrm{OT}(\boldsymbol{x}(t-1), \boldsymbol{x}(t))$. Using the aforementioned Birkhoff-von Neumann theorem, there exists an optimal coupling between $\boldsymbol{x}(t-1) \in \mathcal{P}_k(\mathcal{X})$ and $\boldsymbol{x}(t) \in \mathcal{P}_k(\mathcal{X})$ that takes its values in $\{0, 1/k, \ldots, 1\}$. Concretely, from the collective point of view, this means that we can move the agents from distribution $\boldsymbol{x}(t-1)$ to distribution $\boldsymbol{x}(t)$ using this coupling, without having to split any agent. We use this coupling to reassign the agents deterministically to their new destination (we assume that agents have arbitrary identifiers and the ability to communicate, in order to break the possible ties). Clearly, since the coupling is optimal, the average movement cost in the collective strategy will equal the transport cost of the fractional strategy, and since the population distribution of the agents is consistent with the barely fractional configuration, the service costs are equal. $\qquad \square$

**Remark 2.3** (Fractional metrical task systems, with fixed transaction costs)**.** *A straightforward illustration of the fractional variant of metrical task systems is in asset management. Consider an investor distributing its stakes on several assets via $\boldsymbol{x}(t) \in \mathcal{P}(\mathcal{X})$ where $\mathcal{X}$ is a set of financial products such as stocks. The distance between two products is defined by the transaction cost associated to exchanging a unit value between two product (i.e. the liquidity of the products varies). The service cost corresponds to the cost of holding a poorly performing asset. The movement cost is a variable cost which is proportional to the mass that is exchanged at any round. Another type of cost that could naturally arise is a fixed transaction cost that the player should pay for converting any amount of stock $\rho$ into stock $\rho'$. We propose to set this cost equal to $\tau d(\rho, \rho')$ for some real constant $\tau > 0$. In this setting, it becomes relevant for the player to perform transactions only if they involve a fractional mass greater than $\tau$. A direct consequence of our results (Theorem 1.1 and Proposition 2.2) is that adding fixed transaction cost $\tau$ to the fractional variant of metrical task systems has little effect on its competitive ratio, provided that $\tau \leq 1/n^2$.*

# 3 Potential function method for barely fractional strategies

In this section we provide the proof of the main result of the paper, Theorem 1.1. The core of our method is in Algorithm 1 which provides a way to transform a (black-box) fully fractional strategy $\boldsymbol{y}(\cdot)$ in a $k$-barely fractional strategy $\boldsymbol{x}(\cdot)$, while only loosing a factor 2 in the competitive ratio provided that $k \geq n^2$ (it is then generalised to arbitrary factor $1 + \epsilon$ in the proof below). Following the reductions from Section 2, this method also allows to transform a fully random algorithm into a $k$-barely random algorithm with similar competitive guarantees.

Note that the cost of $\boldsymbol{x}(\cdot)$ must be bounded by a constant times the cost of $\boldsymbol{y}(\cdot)$ if we hope to preserve the competitiveness of the latter. The informal reason why such guarantee is difficult to obtain is that the adversary *knows exactly* the discretization method used by the team (by adversary, we mean the designer of the cost sequence $\boldsymbol{c}(\cdot)$, and by the team, we mean the swarm constituting $\boldsymbol{x}(\cdot)$). For instance, the naive rounding strategy which would be $\boldsymbol{x}(t) \in \arg\min_{\boldsymbol{x} \in \mathcal{P}_k(\mathcal{X})} \mathrm{OT}(\boldsymbol{y}(t), \boldsymbol{x})$ does not preserve competitiveness. This is because infinitesimal movements of $\boldsymbol{y}(\cdot)$ could induce large (order $1/k$) movements of $\boldsymbol{x}(\cdot)$. A good discretization strategy must therefore display a hysteresis phenomenon, in the sense that it does not undo a move right after that move was decided. This is precisely the idea behind our strategy in Equation (1) that employs a potential function $D(\cdot, \cdot)$ to insure that $\boldsymbol{x}(t)$ remains sufficiently close to $\boldsymbol{y}(t)$ and an additional term in $\mathrm{OT}(\boldsymbol{x}(t-1), \boldsymbol{x}(t))$ to limit the movement cost of $\boldsymbol{x}(\cdot)$. We note that the technique presents some similarities with first order optimization methods, which have been successful in several areas of online decision making. One key novelty is that it is applied to a discrete space $\mathcal{P}_k(\mathcal{X})$.

---

**Algorithm 1:** Barely fractional strategy for $k \geq n^2$

---

**Input:** $\boldsymbol{x}(t-1)$: Previous barely fractional configuration
**Input:** $\boldsymbol{y}(t)$: Next fully fractional distribution
**Output:** $\boldsymbol{x}(t)$: Next barely fractional configuration

Solve for $\boldsymbol{x} \in \mathcal{P}_k(\mathcal{X})$:

$$\boldsymbol{x} \in \arg\min_{\boldsymbol{x} \in \mathcal{P}_k(\mathcal{X})} D(\boldsymbol{x}, \boldsymbol{y}(t)) + \mathrm{OT}(\boldsymbol{x}(t-1), \boldsymbol{x}), \tag{1}$$

where $D(\boldsymbol{x}, \boldsymbol{y}) = 2\mathrm{OT}\left(\boldsymbol{x}/2 + 1/(2n)\mathbb{1}, \boldsymbol{y}\right)$, breaking ties by maximizing $\mathrm{OT}(\boldsymbol{x}(t-1), \boldsymbol{x})$.

---

**Theorem 3.1.** *Consider an abritrary $n$ point metric $\mathcal{X}$, some real $\epsilon > 0$ and an integer $k \geq n^2/\epsilon$. For any fractional MTS strategy $\boldsymbol{y}(\cdot)$ on $\mathcal{X}$ which is $\alpha$-competitive, Algorithm 1 with potential defined in* (2) *provides a $k$-barely fractional MTS strategy $\boldsymbol{x}(\cdot)$ on $\mathcal{X}$ which is $(1 + \epsilon)\alpha$-competitive.*

*Proof.* We fix an integer $k \geq n^2/\epsilon$. Consider some $\alpha$-competitive fractional strategy $\boldsymbol{y}(\cdot)$. We shall define (in an online manner) a $(1 + \epsilon)\alpha$-competitive $k$-barely fractional strategy $\boldsymbol{x}(\cdot)$, by performing an appropriate discretization of $\boldsymbol{y}(\cdot)$.

For any pair of distributions $\boldsymbol{x}, \boldsymbol{y} \in \mathcal{P}(\mathcal{X})$ we call the potential between $\boldsymbol{x}$ and $\boldsymbol{y}$ and denote by $D(\boldsymbol{x}, \boldsymbol{y})$ the quantity

$$D(\boldsymbol{x}, \boldsymbol{y}) = (1 + \epsilon)\mathrm{OT}\left(\frac{1}{1 + \epsilon}\boldsymbol{x} + \frac{\epsilon}{n(1 + \epsilon)}\mathbb{1}, \boldsymbol{y}\right), \tag{2}$$

where $\mathbb{1}$ denotes the vector of $\mathbb{R}^{\mathcal{X}}$ with all its values equal to 1. The correctness of this definition stems from the fact that we will always have $\boldsymbol{x}/(1 + \epsilon) + \epsilon/n(1 + \epsilon)\mathbb{1} \in \mathcal{P}(\mathcal{X})$, because $\boldsymbol{x} \in \mathcal{P}(\mathcal{X})$.

Upon arrival of request $\boldsymbol{c}(t) \in \mathbb{R}_+^{\mathcal{X}}$, the (fully) fractional distribution $\boldsymbol{y}(t)$ is computed from a black-box algorithm, and the barely fractional configuration $\boldsymbol{x}(t)$ is defined via Equation (1),

$$\boldsymbol{x}(t) \in \arg\min_{\boldsymbol{x} \in \mathcal{P}_k(\mathcal{X})} D(\boldsymbol{x}, \boldsymbol{y}(t)) + \mathrm{OT}(\boldsymbol{x}(t-1), \boldsymbol{x}),$$

where ties are broken in favor of the configuration $\boldsymbol{x}$ which maximizes $\mathrm{OT}(\boldsymbol{x}(t-1), \boldsymbol{x})$. We now aim to bound the movement and service costs of the aforedefined strategy $\boldsymbol{x}(\cdot)$ in terms of the movement and service costs of the black-box strategy $\boldsymbol{y}(\cdot)$.

*Movement Costs.* We start with the movement cost of $\boldsymbol{x}(\cdot)$. We decompose the change of the potential $P(t) = D(\boldsymbol{x}(t), \boldsymbol{y}(t))$ as follows,

$$
\begin{aligned}
P(t) - P(t-1) &= D(\boldsymbol{x}(t), \boldsymbol{y}(t)) - D(\boldsymbol{x}(t-1), \boldsymbol{y}(t-1)) \\
&= D(\boldsymbol{x}(t), \boldsymbol{y}(t)) - D(\boldsymbol{x}(t-1), \boldsymbol{y}(t)) + D(\boldsymbol{x}(t-1), \boldsymbol{y}(t)) - D(\boldsymbol{x}(t-1), \boldsymbol{y}(t-1)), \\
&\leq -\mathrm{OT}(\boldsymbol{x}(t-1), \boldsymbol{x}(t)) + (1+\epsilon))\mathrm{OT}(\boldsymbol{y}(t-1), \boldsymbol{y}(t)),
\end{aligned}
$$

where the first inequality $D(\boldsymbol{x}(t), \boldsymbol{y}(t)) - D(\boldsymbol{x}(t-1), \boldsymbol{y}(t)) \leq -\mathrm{OT}(\boldsymbol{x}(t-1), \boldsymbol{x}(t))$ comes from the optimality of $\boldsymbol{x}(t)$ in (1) and the other inequality $D(\boldsymbol{x}(t-1), \boldsymbol{y}(t)) - D(\boldsymbol{x}(t-1), \boldsymbol{y}(t-1)) \leq (1+\epsilon)\mathrm{OT}(\boldsymbol{y}(t-1), \boldsymbol{y}(t))$ is a direct application of the triangle inequality for optimal transport. We therefore have the at all times $t \geq 1$,

$$
\mathrm{OT}(\boldsymbol{x}(t-1), \boldsymbol{x}(t)) \leq P(t-1) - P(t) + (1+\epsilon)\mathrm{OT}(\boldsymbol{y}(t-1), \boldsymbol{y}(t)). \tag{3}
$$

Taking the (telescopic) sum with (3), we obtain the desired bound on the movement cost of $\boldsymbol{x}(\cdot)$,

$$
\begin{aligned}
\sum_{t \geq 1} \mathrm{OT}(\boldsymbol{x}(t-1), \boldsymbol{x}(t)) &\leq \sum_{t \geq 1} P(t-1) - P(t) + (1+\epsilon)\mathrm{OT}(\boldsymbol{y}(t-1), \boldsymbol{y}(t)) \\
&\leq P(0) + (1+\epsilon)\sum_{t \geq 1} \mathrm{OT}(\boldsymbol{y}(t-1), \boldsymbol{y}(t)),
\end{aligned}
$$

where $P(0) = D(\boldsymbol{x}(0), \boldsymbol{y}(0))$ is a constant bounded by the diameter of $\mathcal{X}$.

*Service costs.* We shall now show that at all times $t \geq 1$,

$$
\forall \rho \in \mathcal{X} : x_\rho(t) \leq (1+\epsilon)y_\rho(t). \tag{4}
$$

This equation clearly implies that the service cost of $\boldsymbol{x}(\cdot)$ is at most $(1+\epsilon)$ that of $\boldsymbol{y}(\cdot)$. We shall use the following equation from Lemma 3.2 which states that at any time $t$, and for any configuration $\boldsymbol{x}' \in \mathcal{P}_k(\mathcal{X})$ such that $\boldsymbol{x}' \neq \boldsymbol{x}(t)$,

$$
D(\boldsymbol{x}(t), \boldsymbol{y}(t)) < D(\boldsymbol{x}', \boldsymbol{y}(t)) + \mathrm{OT}(\boldsymbol{x}(t), \boldsymbol{x}'). \tag{5}
$$

Assume by contradiction that (4) is not satisfied. For some $\rho \in \mathcal{X}$, we have $\frac{1}{1+\epsilon}x_\rho(t) + \frac{\epsilon}{n(1+\epsilon)} > y_\rho(t) + \frac{\epsilon}{n(1+\epsilon)}$. By applying Lemma 3.3 to the distributions $\mathbf{z} = \frac{1}{1+\epsilon}\boldsymbol{x}(t) + \frac{\epsilon}{n(1+\epsilon)}\mathbb{1}$ and $\boldsymbol{y}(t)$, with $\delta = \frac{1}{k(1+\epsilon)} \leq \frac{\epsilon}{n^2(1+\epsilon)}$, we obtain that there exists some $\rho' \in \mathcal{X}$ such that $\mathbf{z}' = \mathbf{z} + \delta(\boldsymbol{e}_{\rho'} - \boldsymbol{e}_\rho)$ and satisfying $\mathrm{OT}(\mathbf{z}, \boldsymbol{y}(t)) = \mathrm{OT}(\mathbf{z}, \mathbf{z}') + \mathrm{OT}(\mathbf{z}', \boldsymbol{y}(t))$, where $\mathrm{OT}(\mathbf{z}, \mathbf{z}') = \delta d(\rho, \rho')$. We then consider the configuration $\boldsymbol{x}' = \boldsymbol{x}(t) + \frac{1}{k}(\boldsymbol{e}_{\rho'} - \boldsymbol{e}_\rho)$, which will provide the contradiction by forming a violation of (5). First note that $\boldsymbol{x}' \in \mathcal{P}_k(\mathcal{X})$ because $x_\rho(t) \geq 1/k$ since $x_\rho(t) > (1+\epsilon)y_\rho(t) \geq 0$. Also, by definition $\boldsymbol{x}' \neq \boldsymbol{x}(t)$. Finally, $D(\boldsymbol{x}(t), \boldsymbol{y}(t)) = (1+\epsilon)\mathrm{OT}(\mathbf{z}, \boldsymbol{y}(t)) = (1+\epsilon)\mathrm{OT}(\mathbf{z}', \boldsymbol{y}(t)) + (1+\epsilon)\delta d(\rho, \rho') = D(\boldsymbol{x}', \boldsymbol{y}(t)) + \frac{1}{k}d(\rho, \rho') = D(\boldsymbol{x}', \boldsymbol{y}(t)) + \mathrm{OT}(\boldsymbol{x}(t), \boldsymbol{x}')$, which provides the contradiction.

Overall, the movement and service costs of $\boldsymbol{x}(\cdot)$ are bounded by $(1+\epsilon)$ those of $\boldsymbol{y}(\cdot)$ (plus a constant), thus, the $\alpha$-competitiveness of $\boldsymbol{y}(\cdot)$ implies the $(1+\epsilon)\alpha$-competitiveness of $\boldsymbol{x}(\cdot)$. $\qquad\square$

**Lemma 3.2.** *At any time $t$, the configuration $\boldsymbol{x}(t)$ that is selected by Equation* (1) *and the associated tie-breaking rule satisfies that for any $\boldsymbol{x}' \in \mathcal{P}_k(\mathcal{X})$ such that $\boldsymbol{x}' \neq \boldsymbol{x}(t)$,*

$$
D(\boldsymbol{x}(t), \boldsymbol{y}(t)) < D(\boldsymbol{x}', \boldsymbol{y}(t)) + \mathrm{OT}(\boldsymbol{x}(t), \boldsymbol{x}').
$$

*Proof.* Recall that $\boldsymbol{x}(t-1)$ is the configuration which immediately preceded $\boldsymbol{x}(t)$. Consider and some arbitrary configuration $\boldsymbol{x}' \in \mathcal{P}_k(\mathcal{X})$. Since $\boldsymbol{x}(t)$ achieves the minimum of (1), we have

$$
D(\boldsymbol{x}(t), \boldsymbol{y}(t)) + \mathrm{OT}(\boldsymbol{x}(t-1), \boldsymbol{x}(t)) \leq D(\boldsymbol{x}', \boldsymbol{y}(t)) + \mathrm{OT}(\boldsymbol{x}(t-1), \boldsymbol{x}').
$$

Using the triangle inequality of optimal transport, we also have

$$
\mathrm{OT}(\boldsymbol{x}(t-1), \boldsymbol{x}') \leq \mathrm{OT}(\boldsymbol{x}(t-1), \boldsymbol{x}(t)) + \mathrm{OT}(\boldsymbol{x}(t), \boldsymbol{x}').
$$

The combination of both inequalities, proves a non-strict version of Equation (5),

$$
D(\boldsymbol{x}(t), \boldsymbol{y}(t)) \leq D(\boldsymbol{x}', \boldsymbol{y}(t)) + \mathrm{OT}(\boldsymbol{x}(t), \boldsymbol{x}'). \tag{6}
$$

We now explain why this inequality is in fact strict for any $\boldsymbol{x}' \neq \boldsymbol{x}(t)$. We note that equality can be met in (6) only if equality is also met in both of the aforementioned components of the inequality. We assume that is the case for some $\boldsymbol{x}'$ and will now show that $\boldsymbol{x}' = \boldsymbol{x}(t)$. We have that,

$$\begin{cases} D(\boldsymbol{x}(t), \boldsymbol{y}(t)) + \mathrm{OT}(\boldsymbol{x}(t-1), \boldsymbol{x}(t)) = D(\boldsymbol{x}', \boldsymbol{y}(t)) + \mathrm{OT}(\boldsymbol{x}(t-1), \boldsymbol{x}'), \\ \mathrm{OT}(\boldsymbol{x}(t-1), \boldsymbol{x}') = \mathrm{OT}(\boldsymbol{x}(t-1), \boldsymbol{x}(t)) + \mathrm{OT}(\boldsymbol{x}(t), \boldsymbol{x}'). \end{cases}$$

By the tie-breaking rule of (1), the first inequality implies that $\mathrm{OT}(\boldsymbol{x}(t-1), \boldsymbol{x}') \leq \mathrm{OT}(\boldsymbol{x}(t-1), \boldsymbol{x}(t))$ which allows to conclude, using the second equality that $\mathrm{OT}(\boldsymbol{x}(t), \boldsymbol{x}') \leq 0$, and thus that $\boldsymbol{x}' = \boldsymbol{x}(t)$. This relies on the positivity of the optimal transport distance, which follows from the positivity of the distance in the original metric space $\mathcal{X}$. $\qquad\square$

**Lemma 3.3.** *Let* $\mathbf{z} \in \mathcal{P}(\mathcal{X})$ *and* $\boldsymbol{y} \in \mathcal{P}(\mathcal{X})$ *be two distributions on a metric space* $\mathcal{X}$ *with* $n$ *points. Assume that for some real* $\delta > 0$ *and some point* $\rho \in \mathcal{X}$ *one has* $z_\rho \geq y_\rho + n\delta$. *Then, there is another distribution* $\mathbf{z}' \in \mathcal{P}(\mathcal{X})$, *taking the form* $\mathbf{z}' = \mathbf{z} + \delta(\boldsymbol{e}_{\rho'} - \boldsymbol{e}_\rho)$ *for some other point* $\rho' \in \mathcal{X}$, *which satisfies* $\mathrm{OT}(\mathbf{z}, \boldsymbol{y}) = \mathrm{OT}(\mathbf{z}, \mathbf{z}') + \mathrm{OT}(\mathbf{z}', \boldsymbol{y})$, *where* $\mathrm{OT}(\mathbf{z}, \mathbf{z}') = \delta d(\rho, \rho')$.

*Proof.* By the triangle inequality of the transport distance, for any three distributions $\mathbf{z}, \mathbf{z}', \boldsymbol{y} \in \mathcal{P}(\mathcal{X})$, we have $\mathrm{OT}(\mathbf{z}, \boldsymbol{y}) \leq \mathrm{OT}(\mathbf{z}, \mathbf{z}') + \mathrm{OT}(\mathbf{z}', \boldsymbol{y})$, therefore it will suffice to show the converse inequality.

We consider an optimal coupling from $\mathbf{z}$ to $\boldsymbol{y}$ denoted by $\pi = (\pi(\rho_1, \rho_2))_{\rho_1, \rho_2 \in \mathcal{X}^2}$. By definition, $z_\rho - y_\rho = \sum_{\rho_1 \in \mathcal{X}} \pi(\rho, \rho_1) - \pi(\rho_1, \rho) \geq n\delta$. In this sum of $n$ terms, at least one term has to exceed $\delta$. This allows to define one point $\rho' \in \mathcal{X}$ such that $\pi(\rho, \rho') \geq \delta$.

Next we consider the distribution $\mathbf{z}' = \mathbf{z} + \delta(\boldsymbol{e}_{\rho'} - \boldsymbol{e}_\rho)$. First, observe that this distribution is well-defined, since $z_\rho \geq \delta$. Then, we note that $\pi' = \pi + \delta(\boldsymbol{e}_{\rho'} \otimes \boldsymbol{e}_{\rho'} - \boldsymbol{e}_\rho \otimes \boldsymbol{e}_{\rho'})$ defines a coupling from $\mathbf{z}'$ to $\boldsymbol{y}$ and thus,

$$\mathrm{OT}(\mathbf{z}', \boldsymbol{y}) \leq \sum_{\rho_1, \rho_2} d(\rho_1, \rho_2) \pi'(\rho_1, \rho_2) = \mathrm{OT}(\mathbf{z}, \boldsymbol{y}) - \mathrm{OT}(\mathbf{z}, \mathbf{z}'),$$

which finishes the proof. $\qquad\square$

# 4 Discussion, lower-bound and open directions

## 4.1 Uniform metric space

The uniform metric space is is a notoriously simple special case of metrical task systems. It is the metric space where all pairs of nodes are at the same distance $\forall \rho, \rho' \in \mathcal{X}, \rho \neq \rho' \implies d(\rho, \rho') = 1$. The (fully) randomized competitive ratio of metrical task systems on the uniform metric space is known to lie between $H_n$ and $2H_n$ [12], where $H_n$ is the $n$-th harmonic number.

In this section, we show that $\lceil \log n \rceil$ bits of randomness are sufficient to achieve a constant factor optimal competitive ratio on the uniform metric space. The proof can be seen as a direct discretization of the algorithm of [12], and as an application of the 'game of balls and urns' in [25].

**Proposition 4.1.** *There is a* $n$-*barely random and* $2H_n + 6$-*competitive algorithm for metrical task systems on the uniform metric space of cardinality* $n$.

*Proof.* Following [12], we decompose the time in discrete phases, in order to obtain a lower bound on $\mathrm{OPT}(\boldsymbol{c}(\cdot))$. The 0-th phase starts at $t_0 = 0$ and for $i \in \mathbb{N}$, the $r$-th phase begins at time $t_i$ and ends right before $t_{i+1}$ which is defined as the first instant satisfying that $\forall \rho \in \mathcal{X} : \sum_{t_i \leq t < t_{i+1}} c_\rho(t) \geq 1$. Without loss of generality we can assume for simplicity that there always exist a state $\rho_i$ such that $\sum_{t_i \leq t < t_{i+1}} c_{\rho_i}(t) = 1$ (see the continuous-time variant of metrical task systems, as in [14], for a justification). We denote by $i^*$ the total number of phases, which only depends on the cost sequence $\boldsymbol{c}(\cdot)$. We observe that $\mathrm{OPT}(\boldsymbol{c}(\cdot)) \geq i^*$, because the cost incurred by any fixed trajectory is at least one during a given phase, and the cost of any mobile trajectory is also at least one.

The idea of [12] is to define a randomized algorithm that incurs a cost of at most $2H_n$ in each phase, and which is thus $2H_n$-competitive. In a given phase $i$, for any $t \in [t_i, t_{i+1})$, the strategy consists in assigning $\boldsymbol{x}(t)$ to be the uniform probability distribution over the states in $\{\rho \in \mathcal{X} : \sum_{t_i \leq \tau \leq t} c_\rho(\tau) \leq 1\}$, which are the non-saturated states. This strategy suffers a movement cost of

at most $1 + 1/n + 1/(n-1) \cdots + 1/2 = H_n$, where $H_n$ denotes the $n$-th harmonic number. Note that the service cost of this strategy is less or equal to its movement cost, so it is $2H_n$-competitive. Ufortunately, the strategy is not $n$-barely fractional, because it takes values in $\{1/n, 1/(n-1), \ldots, 1\}$ which are not all multiples of $1/n$.

We consider a simple discretization of this strategy, which does not suffers from the caveat described in Section 3. This descritization relies on the decomposition in phases which is specific to the uniform metric space. It can be seen as an application of the analysis of the 'game of balls and urns' which was used to design a collective tree exploration algorithm in [25]. We recall the definition of the game. The game starts with $n$ balls each located in one of $n$ urns. In discrete rounds, an adversary picks an urn which contains a ball and the player has to move one ball in that urn to any other urn. The game ends when all urns have been chosen at least once by the adversary, and the cost of the player is the number of rounds before the end of the game, re-normalized by $1/n$. The simple strategy for the player that consists in relocating selected balls to the least loaded urn is known to induce a cost of at most $\ln(n) + 3 \leq H_n + 3$. Observe that this game models exactly the movement cost a corresponding $n$-barely fractional strategy in a given phase (where each ball represents a probability mass of $1/n$ and each urn represents a point of the uniform metric space). Also, since the service cost is (again) bounded by the movement cost in a given phase, the strategy is $2H_n + 6$-competitive. $\quad\square$

### 4.2 Lower-bound on the number of random bits

We now explain why $\Omega(\log n)$ random bits are required to obtain an asymptotically optimal competitive ratio. We show the stronger result that a $k$-barely random algorithm for metrical task systems is at most $(2n-1)/k$-competitive. Thus, a $\mathcal{O}(\log^2 n)$-competitive strategy must have $k \geq cst \times n/\log^2 n$ – corresponding to a lower-bound of $\log k \geq \log n - \mathcal{O}(\log \log n)$ in the number of random bits.

**Proposition 4.2.** *The competitive ratio of $k$-barely random algorithms for metrical task systems is at least $(2n-1)/k$.*

*Proof.* Consider a barely random algorithm $\mathcal{A}(\cdot, \boldsymbol{s})$, with $\boldsymbol{s} \sim \mathcal{D} = \mathcal{U}(\{1, \ldots, k\})$. The deterministic algorithm $\mathcal{A}(\cdot, 1)$ is sampled with probability $1/k$. By the $2n-1$ lower-bound of [12] on the competitive ratio of deterministic algorithms for metrical task systems, there exists a sequence $\boldsymbol{c}(\cdot) \in \mathbb{R}_+^{\mathbb{N}}$ such that $\mathrm{Cost}(\boldsymbol{\rho}^{\mathcal{A}(\cdot, 1)}(\cdot), \boldsymbol{c}(\cdot)) \geq (2n-1)\mathrm{OPT}(\boldsymbol{c}(\cdot))$. Integrating over the seed $\boldsymbol{s} \sim \mathcal{U}(\{1, \ldots, k\})$, we get $\mathbb{E}_{\boldsymbol{s}}(\mathrm{Cost}(\boldsymbol{\rho}^{\mathcal{A}(\cdot, \boldsymbol{s})}(\cdot), \boldsymbol{c}(\cdot))) \geq \frac{2n-1}{k}\mathrm{OPT}(\boldsymbol{c}(\cdot))$. $\quad\square$

The question of whether an order-optimal randomized competitive ratio can be achieved with $k = O(n)$ instead of $k \geq n^2$ remains open on general metric spaces. This seems plausible for weighted star metrics, which admit a $\mathcal{O}(\log n)$ competitive strategies via an algorithm working in phases [17]. It is also possible that some metric spaces do require $k = \Omega(n^2)$. Demonstrating such cutoff for barely random algorithms would nicely echo the cutoff between $\Theta(\log n)$ and $\Theta(\log^2 n)$ that seems to appear in the competitive ratio.

### 4.3 Conclusion and open directions for collective algorithms

The term 'collective algorithm' used in this paper is borrowed from the literature on distributed algorithms [34, 24]. We believe that highlighting situations in which a team of agent can be competitive/intelligent in a way that no single agent could be is an interesting research direction. As presented in this paper, such situations echo the well-studied competitive gap between randomized and deterministic algorithms for online problems [7]. But collective algorithms come with considerations that go beyond their analogy with barely-random algorithms. For instance, one can study the competitiveness of collective algorithms under: restricted communications (such as the write-read model of [34] or the local model of [27]), synchronous movements (as discussed in [23]), capacity constraints (where the amount of agents at a position is limited [6, 25]) or even with agents having some degree of selfishness or maliciousness (then involving a price of anarchy [39] for selfish agents, or some degree of robustness to byzantine faults [40] for malicious agents).

**Acknowledgements** The authors thank the anonymous reviewers for their helpful comments. RC thanks Spyros Angelopoulos, Nikhil Bansal, Christian Coester and Christoph Dürr for discussions. This work was supported by PRAIRIE ANR-19-P3IA-0001.

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
