# OpenReview forum: "Barely Random Algorithms and Collective Metrical Task Systems"
_NeurIPS.cc/2024/Conference — NeurIPS 2024 spotlight_

### Official Review · Reviewer_1Eyx · 2024-07-11

**Soundness:** 4
**Presentation:** 4
**Contribution:** 3
**Rating:** 6
**Confidence:** 3

**Summary:**

This paper introduces and studies the problem of designing and analyzing randomized algorithms for Metrical Task Systems (MTS) using only limited randomness, that is, a number of random bits that only depends on $n$ (the number of states in the MTS), rather than the length of the sequence, as in the case of the known SOTA randomized algorithms. The main result of the paper shows that $2\log n$ random bits suffice to achieve the same competitive ratio as the best fully randomized algorithms, up to a factor of 2. This is near-optimal, as shown, in the sense that $\log n-O(\log \log n)$ bits are necessary to this end. The barely-random setting with $k$ random bits also yields directly a solution to a {\em collective} MTS setting, in which, informally, $k$ algorithms are run in parallel, and the cost paid is the average of the cost of the $k$ algorithms. This collective setting is motivated by similar in spirit works on graph exploration in distributed settings, with $k$ parallel agents.  The approach builds on a fractional formulation of the online problem, and techniques from optimal transport theory, such as the Birkhoff-von Neuman theorem, instead of the standard tree embedding techniques.

**Strengths:**

The paper is very well written. All results are rigorously proven, but there is also a lot of intuition and high-level guiding of the reader.
The solution is non-trivial and builds on fairly complex ideas.  Furthermore, even though the work is theoretical, it is connected to concrete applications.

The idea of using concepts from optimal transport theory looks novel, although I am not familiar with its application to other online problems via competitive analysis.

**Weaknesses:**

There is no experimental analysis. It is understandable that the work is mainly theoretical, but one suggestion would be to expand on the application discussed in Remark 2.3 and include an experimental evaluation of this problem.

**Questions:**

Is the use of optimal transport theorem novel in the competitive analysis of MTS problems?

For the setting of Remark 2.3, is this application known in the literature, or you define it?

In Section 3 it would be good to have the statement of the algorithm in rough pseudocode.

line 400: Can you expand on these similarities with descent methods?


Other comments:

lines 141-148: please add appropriate citations throughout.

I assume the reason behind the proof of Proposition 2.1 is to give intuition about the more complex setting? If yes please make this clear.

line 203: typo "achieves".

line 280: This is the potential function, but it is hidden, please make it more prominent.

**Limitations:**

No issues in regards to limitations.

---

> ### Author Rebuttal · Authors · 2024-07-31
>
> We thank the Reviewer for their interest and their thoughtful questions.
>
> * To the best of our knowledge, the use of the Birkhoff-von Neumann theorem in the context of competitive analysis is new. We build upon this theorem in Proposition 2.2 of Section 2.2. As the reviewer points out, the purpose of Proposition 2.1 (which is well-known in the literature) is just to give more intuition on the more complex setting. We will highlight this in the revised version of the paper. The other key novel technical part of our analysis is in Section 3 where we develop a first-order method that displays a *hysteresis phenomenon* to track a fully fractional configuration with a barely fractional configuration. This shows the existence of $O(\log^2 n)$-competitive barely fractional configurations, which in turn has implications for randomized algorithms, collective algorithms and advice complexity.
>
> * The application suggested in Remark 2.3 is quite natural: since our algorithm only moves significant amounts of mass, it is not sensitive to (small) fixed transaction costs ; this comes in contrast with fully fractional algorithms, because they move arbitrarily small amounts of mass. We believe that this application is also new and was out of reach for previous methods.
>
> * As suggested by the reviewer, we will add a (very short) pseudocode to highlight the simplicity of our method.
>
> * Section 3 is reminiscent of offline first order optimization in many ways. Gradient descent of a function $f(\cdot)$ is also known as the explicit Euler method, where one can write the $t+1$-th iterate of the method as the minimizer of the sum of the first order approximation of $f(\cdot)$ at the previous iterate and a quadratic cost, i.e.
> \begin{align} x(t+1) = \arg\min_x \nabla f(x(t))(x-x(t)) + ||x-x(t)||^2.
> \end{align}
> A very closely related method is the implicit Euler method where the $t+1$-th iterate is the minimizer of the sum of $f(\cdot)$ and a quadratic cost,
> \begin{align}
> x(t+1) = \arg\min_x f(x) +||x-x(t)||^2,
> \end{align}
> which is also known as the Proximal operator. Our Equation (2) is reminiscent of this Proximal operator, which has important stability properties. One key difference is that we replace the squared norm by an optimal transport cost, and that the offline objective function $f(\cdot)$ is replaced by a potential which is defined online. Note that quite recently, the notion of Wasserstein proximal operator has also appeared as a central object of study in the context of diffusion and partial differential equations (see e.g. the review of Santambrogio on Gradient Flows). A key difference with our Equation (2) is that we consider the Wasserstein-1 metric (aka, the earthmover distance) and not the Wasserstein-2 metric. We will add a discussion on the reviewer's question in the final version of the paper.
>
> * If the reviewer is satisfied with the novelty of our method, and with the proposed revisions, we kindly encourage them to positively reassess their rating of the submission.
>
> * We will correct all typos, put more emphasis on the potential function and add citations in lines 141-148.

---

> > ### Comment · Reviewer_1Eyx · 2024-08-11
> >
> > Dear authors, thank you for your helpful response. I would suggest to include your comment on Section 3 in the paper, if possible,

---

### Official Review · Reviewer_Kc9y · 2024-07-12

**Soundness:** 4
**Presentation:** 3
**Contribution:** 3
**Rating:** 6
**Confidence:** 3

**Summary:**

The current paper addresses the problem of metrical task systems on a general space with $n$ points and proposes a technique to reduce the amount of randomness used by any random algorithm. More precisely, the authors prove that any "fully random" algorithm, which uses an unlimited number of random bits, can be transformed into a "barely random" algorithm that uses only $\lceil 2 \log n \rceil$ random bits, with a competitive ratio at most twice as large. An immediate consequence of this result, using previous works, is the existence of an $O((\log n)^2)$-competitive algorithm that uses only $\lceil 2 \log n \rceil$ random bits. The authors also show that any $O((\log n)^2)$-competitive algorithm requires at least $\log n - O(\log \log n)$ random bits, essentially proving the tightness of their main result.

**Strengths:**

* The results of the paper are interesting and may inspire future work in similar directions.
* Viewing the problem as a collaborative MTS with $k$ deterministic agents provides better insights and understanding of the problem.
* The paper introduces new analysis techniques that could have broader applications.

**Weaknesses:**

I don't see any major weaknesses in the presented results or the proofs.

However, I find that the paper is not a good fit for NeurIPS at all. A conference or journal focused on theoretical computer science would be a much better venue for this work and would provide it with greater visibility among researchers who are more likely to benefit from it.

**Questions:**

Do the authors think that some arguments and techniques from the paper can be used to prove similar results on barely random algorithms in other problems in competitive analysis?

**Limitations:**

The assumptions of the theoretical results are clearly stated.

---

> ### Author Rebuttal · Authors · 2024-07-31
>
> We thank the reviewer for their interest and their careful reading. We note that the reviewer's appreciation of our results seems very positive, and that their only concern is whether the paper is a good fit for NeurIPS. We attempt to convince the reviewer that this is indeed the case, in the hope that they would increase their rating in a way that reflects their appreciation of the paper's inherent quality.
>
> While metrical task systems (MTS) is a problem that originated in the 1990s from the community of theoretical computer science (TCS), we observe that in the last 5 years the problem has gained a strong interest in the machine learning community, with many contributions on *learning augmented algorithms* [1,17,2,18,3] and on *concrete applications* of MTS [37, 25]. Most of these papers were published at NeurIPS.
>
> We note that our work has a flavor that is very similar to *learning-augmented algorithms*. Learning-augmented algorithms use predictions coming from a black-box algorithm, in the hope to improve over the performance of competitive algorithms when predictions are accurate. In a similar fashion, our method tracks a fully fractional configuration seen as coming from a black-box algorithm, and makes it barely fractional in a competitive way. We therefore expect our methods could be relevant to researchers working on learning-augmented algorithms, for instance when they need to interpolate smoothly between black-box configurations coming from one or many predictors.
>
> Our work is also motivated by *concrete applications*. The method we propose is very easy to implement and can be applied to concrete problems such as dynamic power management [2], distributed systems [30,21] or asset management (Remark 2.3). For a very immediate example of an application of our work to a power management problem, we encourage the reviewer to read the second item of our response to **Reviewer stvh**.
>
> The result on the existence of a barely random algorithm for MTS has indeed a theoretical flavor. But it is only one of the consequences of our analysis. Our results also imply for e.g. that a deterministic (finite) team of agents can be competitive in a way no single agent can be – which echoes of the study of intelligence in multi-agent systems. A totally different interpretation of our results is in advice complexity (Corollary 1.4) a subject which is very connected to learning-augmented algorithms. For more details on this aspect of the paper, we encourage the reviewer to read the first item of our response to **Reviewer stvh**.
>
> Before submitting this paper to NeurIPS, we carefully verified the "NeurIPS 2024 Call for Paper" which lists the topics in the scope of the conference. We felt that our paper falls entirely within the scope described by this document (specifically, at the intersection of Theory, Optimization and Online Machine Learning). We encourage the reviewer to check these guidelines when finalizing their rating.
>
> Finally, we understand and truly appreciate the intent of the Reviewer to make the results of the paper known to the TCS community. We will not miss other opportunities (workshops, seminars or other submissions) to engage with this community as well, in the hope to develop application-driven and theoretically-grounded works for online settings.
>
> * Regarding the reviewer’s question, we believe that our techniques could help to prove new results for other online problems (such as k-servers or metrical service systems), even though there is no immediate reduction.

---

> > ### Comment · Reviewer_Kc9y · 2024-08-07
> > **I raised my score**
> >
> > I thank the authors for their response.
> >
> > I am very familiar with the fields of advice complexity and learning-augmented algorithms, and aware of previous works on MTS and online algorithms published in ML venues.
> >
> > I agree that the paper has strong connections to the field of advice complexity, which is somewhat related to learning-augmented algorithms. However, the core aspect of the latter framework, which I believe makes it suitable for ML venues, is the necessity of dealing with potentially inaccurate predictions. This aspect is not addressed in studies of advice complexity, which are out of scope for NeurIPS.
> >
> > That said, since the other reviewers do not see this as an issue, and because I mostly liked the paper and find its contributions very interesting (as noted in my review), I will change my score and join the other reviewers in recommending acceptance.

---

### Official Review · Reviewer_stvh · 2024-07-12

**Soundness:** 3
**Presentation:** 3
**Contribution:** 3
**Rating:** 7
**Confidence:** 4

**Summary:**

Authors study randomized algorithms for Metrical Task Systems (MTS)
which need only a small number of random bits which, in particular,
does not depend on the length of the time horizon.
They also interpret this in terms of an average performance of
a cooperating group of several deterministic algorithms
and in terms of advice complexity of reaching the performance
of the best randomized algorithms for MTS using a deterministic algorithm.
Their main result is a reduction which can turn any alpha-competitive
randomized algorithm for MTS into a 2alpha-competitive algorithm
which uses only 2*log n random bits.

**Strengths:**

* Proposed results are very interesting. I even find them surprising.

* MTS is one of the central problems in online algorithms. It is extensively
studied in terms of randomized and deterministic algrithms as well as advice
complexity. The paper brings interesting and novel point to the literature on MTS.

* The proposed result connects several distinct views of the problem:
collective algorithms, barely random algorith, advice complexity.
I appreciate the way authors explain this context.

* presentation of the results is clean and elegant.

**Weaknesses:**

* No upper bounds for $k < n^2$. This is a mild weakness, I consider the
presented results strong enough.

**Questions:**

* advice complexity of MTS was already studied, you may want to add missing references,
e.g. Emek et al. Would be helpful for the reader to compare the regime of their results to yours.

* at lines 82,83, there is a repeated word "tight"

* Acquiring random bits sometimes limits usability of randomized algorithms. Caching would be one
example. Can you give an example of a concrete MTS where your method could ease
implementation of randomized algorithms?

**Limitations:**

Explained well in statements of the results.

---

> ### Author Rebuttal · Authors · 2024-07-31
>
> We thank the reviewer for their constructive feedback and suggestions. We answer the reviewer's questions below.
>
> * The reference "Online Computation with Advice" by Emek et al. pointed out by the reviewer is very relevant and will be added to the paper: Emek et al. study the situation where $b$ bits of advice are provided at each time-step to a deterministic MTS algorithm, for $b$ of order $\log n$. The total amount of advice in this reference is therefore of order $B = \Theta(\log n*T)$, where $T$ is the number of time-steps. One surprising consequence of our results (Corollary 1.4) is that we present a deterministic MTS algorithm that has a competitive ratio of $O(\log^2 n)$ using a *single* advice of total size $B = 2\log n$, i.e. $B$ is now independent of the number of time-steps $T$. As suggested by the reviewer, we will add a brief discussion on this matter in the revised version of the paper.
>
> * As the reviewer points out, we also provide a randomized algorithm which requires very few random bits (again, $B= 2\log n$, independently of the input size). The method is therefore helpful in any setting where acquiring random bits is costly. In concrete implementations, we also expect that barely fractional configurations (which are discrete objects) are generally much more practical to handle for computers than fully fractional configurations (which are arbitrary real numbers). But perhaps the most applied consequence of our results is that a deterministic team of k agents can be competitive in a way that no single agent could be (provided that $k\geq n^2$). This has for instance a very direct implication in energy management [2]. Consider a computer with three energy modes (*on*, *sleep*, *off*) with a switching cost of $1$ between *on* and *sleep* and another switching cost of (say) $5$ between *sleep* and *off*. At each time $t$ the request is either that
>    * the computer is used: in which case $c(t) = (1, +\infty, +\infty)$ so the computer must be in state *on* ; or
>    * the computer is not used: in which case $c(t) = (1, 0.5, 0)$ so the computer would rather be *off* than *sleep*, and *sleep* than *on*.
>
>   This simple setting was presented by S. Bubeck to motivate the relevance of MTS in a series of lectures in 2019 (Five miracles of mirror descents, available on Youtube). Consider the variant of the problem, where the computer is in fact composed of $k = 9$ components (e.g. screen, CPU cores, etc.) that can each individually switch between one of the three modes (*on*, *sleep*, *off*). The energy spent by the computer is the sum of energy spent by its constituents. Since $n=3$ and $k > n^2$, our results imply that this system can enjoy the randomized competitive ratio (up to a factor 2) against a deterministic adversary!
>
> * Typos will be corrected.

---

> > ### Comment · Reviewer_stvh · 2024-08-09
> >
> > thank you for your explanation.

---

### Official Review · Reviewer_DDub · 2024-07-13

**Soundness:** 4
**Presentation:** 4
**Contribution:** 4
**Rating:** 8
**Confidence:** 5

**Summary:**

The paper bounds the randomness required to achieve the asymptotically tight randomized competitive ratio for metrical task systems, a fundamental model of online computing (essentially, prediction with expert advice endowed with a metric cost function for switching among experts). It shows that $2\log n$ random bits (where $n$ is the number of states/experts of the system) are sufficient to achieve a competitive ratio that is within a factor of $2$ of the optimal ratio (which is known in general to be $\Theta(\log^2 n)$, though for some metrics $O(\log n)$ is achievable). It also shows a nearly matching lower bound of $\log n$ bits. The main result, the upper bound, is a rather simple and elegant reduction that transforms any randomized competitive algorithm to a ``barely fractional" algorithm (which uses a finite set of probabilities that are all multiples of $1/n^2$), which in turn implies a ``barely randomized" algorithm. The reduction is not trivial as simple rounding to the nearest multiple of $1/n^2$ probability vector gives very poor performance. Hence, a more sophisticated rounding scheme is employed, where instead of optimizing the distance between the fully randomized algorithm's probabilistic position to the rounded position, one factors in a smoothed version of the rounded position (averaging between it and a uniform distribution), and the switching cost of the rounded algorithm.

**Strengths:**

Metrical task systems are a widely acknowledged fundamental model of online computing. The question is a fundamental question in online computing. The result answers a fundamental question in online computing. It show in particular that the amount of randomness needed is independent of the input sequence length. This has implications to advise assisted online computing, and to multi agent online computing.

**Weaknesses:**

It's a simple idea and a simple proof, though that's not necessarily a weakness.

**Questions:**

None.

**Limitations:**

None.

---

> ### Author Rebuttal · Authors · 2024-07-31
>
> We thank the reviewer for their encouraging review!

---

### Author Rebuttal · Authors · 2024-07-31

We thank the reviewers for their interest, their careful reading, and their feedback to improve the paper.

We quote all four reviewers who unanimously appreciated the content and the presentation of the paper:
* **Reviewer Ddub**: "The result answers a fundamental question in online computing."
* **Reviewer stvh**: "Proposed results are very interesting. I even find them surprising." "Presentation of the results is clean and elegant."
* **Reviewer 1Eyx**:  "The paper is very well written." "The solution is non-trivial and builds on fairly complex ideas."
* **Reviewer Kc9cy**: "The paper introduces new analysis techniques that could have broader applications."

**Reviewer Kc9cy**, whom we thank for their valuable point of view, suggested borderline rejection not on the foundation of the paper’s merits, but because they question its relevance to the publishing venue. Specifically, the reviewer says that the paper would have more impact if it were published in the computer science community. We reply in detail to **Reviewer Kc9cy**, explaining why the present paper can appeal to many researchers attending NeurIPS 2024 and, more broadly, to the online machine learning community.

We also answer the questions of **Reviewer stvh** and **Reviewer 1Eyx** in the hope of alleviating any remaining doubts.

---

### Decision · Program_Chairs · 2024-09-25

**Decision:**

Accept (spotlight)

**Comment:**

This paper presents fundamentally new ideas for an important and well-studied problem.  The reviewers particularly found the ideas to be accessible and understandable.  Together, this makes for a paper with good deep results that are accessible to the community, despite their technical depth.